# Work-Up and Outcome of Hepatic Resection for Peri-Hilar Cholangiocarcinoma (PH-CCA) without Staging Laparoscopy

**DOI:** 10.3390/cancers14071841

**Published:** 2022-04-06

**Authors:** Santhalingam Jegatheeswaran, Panagiotis Stathakis, Harry V. M. Spiers, Fawwaz Mohammed, Panagiotis Petras, Thomas Satyadas, Michael J. Parker, Angela Lamarca, Saurabh Jamdar, Aali J. Sheen, Ajith K. Siriwardena

**Affiliations:** 1Regional Hepato-Pancreato-Biliary Surgical Unit, Manchester Royal Infirmary, Manchester M13 9WL, UK; santhalingam.jegatheeswaran@mft.nhs.uk (S.J.); panos.stathakis@mft.nhs.uk (P.S.); hs781@cam.ac.uk (H.V.M.S.); fawwaz.mohammed@mft.nhs.uk (F.M.); panagiotis.petras@mft.nhs.uk (P.P.); thomas.satyadas@mft.nhs.uk (T.S.); saurabh.jamdar@mft.nhs.uk (S.J.); aali.sheen@mft.nhs.uk (A.J.S.); 2Critical Care Medicine, Manchester Royal Infirmary, Manchester M13 9WL, UK; michael.parker@mft.nhs.uk; 3Christie Hospital, Palatine Road, Manchester M20 4BX, UK; angela.lamarca@nhs.net; 4Faculty of Biology, Medicine and Health, University of Manchester, Manchester M13 9PL, UK; 5Centre of Biosciences, Manchester Metropolitan University, Manchester M15 6BH, UK

**Keywords:** peri-hilar cholangiocarcinoma, surgery, laparoscopy

## Abstract

**Simple Summary:**

This is a single centre cohort of patients undergoing surgery for PH-CCA suggests that routine staging laparoscopy may not be necessary in the pre-operative work-up.

**Abstract:**

**Background:** This study reports the outcome of a work-up programme for resection of peri-hilar cholangiocarcinoma (PH-CCA) without the use of staging laparoscopy. **Methods:** This is a clinical case cohort series of patients undergoing surgical resection of PH-CCA without the use of staging laparoscopy in the work-up algorithm. During the 13 years from 1 January 2009 to 1 January 2022, 32 patients underwent laparotomy for planned surgical resection of PH-CCA. Data were collected on demographic profile, admission biochemistry, radiology, pre-operative intervention, operation and outcome, together with post-operative complications and disease-free and overall survival. **Results:** All patients underwent pre-operative contrast-enhanced CT. Twenty-four (75%) underwent pre-operative MR. Twenty-three (72%) underwent pre-operative biliary drainage. Twenty-nine patients (91%) had either type III or IV peri-hilar cholangiocarcinoma. One patient (3%) in this series underwent a non-resectional laparotomy. Twenty-nine (91%) had a final histopathological diagnosis of PH-CCA. One further patient had a final diagnosis of an intraductal papillary neoplasm of the biliary tree (IPNB) with high-grade dysplasia but no invasive cancer. Eleven patients (36%) received chemotherapy after surgery. The median (95% CI) time to recurrence was 14 (7–31) months. The median survival was 25 (18-upper limit not reached) months. **Conclusion:** This cohort of 32 patients undergoing attempted resection for PH-CCA without the use of staging laparoscopy in the work-up algorithm indicates that with careful attention to patient fitness and cross-sectional and interventional radiologic/endoscopic imaging, a very low non-therapeutic laparotomy rate of 3% can be achieved and sustained.

## 1. Introduction

Peri-hilar cholangiocarcinoma (PH-CCA) is a rare tumour of the proximal biliary confluence [1]. PH-CCA is thought to originate from large-duct type cells in the biliary lining epithelium and peri-biliary glands and shows a high frequency of mutation in KRAS and TP53 genes [2,3]. PH-CCA has a tendency for peri-neural infiltration beyond the macroscopic tumour and the involvement of hilar nodes [1]. Surgical resection of the tumour followed by adjuvant chemotherapy is regarded as the current optimal treatment; however, this is only feasible in a minority of patients with PH-CCA [4]. Typical reasons for non-resectability include patient factors such as co-morbidity and frailty, oncological factors such as either the presence of metastases and/or locally advanced tumour or combinations of both [4,5].

Modern high-resolution cross-sectional imaging in the form of either contrast-enhanced computed tomography (CT), magnetic resonance scan (MR) or both will provide accurate information on local resectability and on the presence or absence of metastases [1,4]. Additional information may also be gained from the use of ^18^fluoro-deoxy-glucose positron emission scanning [6]. The critical factor governing resectability is typically the involvement of the vasculature in the future remnant liver. It is difficult to accurately assess this vascular involvement with laparoscopy. There is variation in the published literature around the indications for staging laparoscopy, with some reports mandating routine use [7] and others recommending more selective use [8]. Staging laparoscopy is operator-dependent, and in a comprehensive systematic review had a range of sensitivity to detect unresectable disease from 31% to 75%, with a pooled sensitivity of 52.2% (95% confidence interval 47.1–57.2%) [9]. This paper reports the outcome of a policy of case selection for surgical resection for patients with PH-CCA based on an algorithm of the assessment of patient fitness, the assessment of tumour and the presence or absence of metastases using cross-sectional imaging, together with additional information on tumour involvement of the biliary tree, obtained either with percutaneous trans-hepatic cholangiography (PTC) and/or endoscopic retrograde cholangiography (ERCP). Staging laparoscopy was not utilised in this work-up protocol. Thus, this study reports the outcome of a work-up programme of surgery for peri-hilar cholangiocarcinoma (PH-CCA) without prior staging laparoscopy.

## 2. Methods

### 2.1. Design

This is a clinical case cohort series of patients undergoing surgical resection of PH-CCA without the use of staging laparoscopy in the work-up pathway.

### 2.2. Setting

The study was undertaken in the Greater Manchester regional Hepato-Pancreato-Biliary (HPB) service. All HPB resectional surgery was undertaken at the Manchester Royal Infirmary. All medical and clinical oncological treatment was undertaken at the Christie Hospital, Manchester. Together, this tertiary HPB centre serves a population of 3.2 million people.

### 2.3. Patients

All patients in this study were under the care of an individual consultant HPB surgeon (A.K.S.). The study period consisted of the 13 years from 1 January 2009 to 1 January 2022.

#### 2.3.1. Inclusion Criteria

Patients were included if they underwent surgical resection of PH-CCA. Patients who were scheduled to undergo resection but in whom operative findings resulted in no resectional procedure were also included.

#### 2.3.2. Exclusion Criteria

Patients undergoing hepatectomy for intra-hepatic cholangiocarcinoma or gallbladder carcinoma were excluded. Patients undergoing hepatectomy with excision of the extra-hepatic biliary tree for non-PH-CCA indications such as complex recurrent colorectal liver metastases were also excluded. Patients were also excluded if they underwent planned segment III bypass.

Patients were also excluded if they had extra-hepatic metastatic disease, hepatic metastases, tumour involvement of the arterial or portal inflow to the future remnant liver or an Eastern Co-operative Oncology Group (ECOG) performance status of >2 [10].

Thirty-one patients underwent surgical resection of PH-CCA during this study period. One further patient with a clinical diagnosis of a resectable PH-CCA underwent a non-therapeutic laparotomy. These thirty-two patients constitute the study population categorised by intention to undergo resection of PH-CCA. During the period of the study, four hundred and eighty-seven hepatic resections were carried out by the individual HPB surgeon (A.K.S.). One hundred and seventy-six (36%) of these were classified as major (hemi-liver or more extensive) [11].

### 2.4. Data Collection

Data were collected on a range of variables following the treatment course, in compliance with the STROBE Statement checklist [12].

#### 2.4.1. Demographic Profile and Admission Biochemistry

Data were recorded prospectively on demographic profile (age, gender and date of operation), admission biochemistry (admission bilirubin, bilirubin at time of surgery and time to normalisation in those patients who presented with raised levels) and time course of treatment (delay from index hospital admission to surgery). For the purposes of this study, the index hospital admission was regarded as the first admission to this hospital when either the diagnosis was made or the patient was admitted for treatment of hyperbilirubinaemia.

#### 2.4.2. Radiologic Findings

Data were collected on diagnostic tests undertaken prior to surgery including CT, MR, ^18^fluro-deoxy-glucose positron emission tomography (FDG-PET), PTC and ERCP. In relation to PTC, data were recorded on unilateral and bilateral drainage and the number of drainage episodes. For ERCP, a record was made of the number of endoscopic interventions and whether a combined percutaneous-endoscopic intervention was utilised.

#### 2.4.3. Operative Detail

Data were collected on operative findings, the detail and extent of the operative procedure including type of resection and whether there was concomitant arterial or venous resection. The use of a frozen section was also recorded. The Tokyo update of the Brisbane nomenclature of liver resection was used, although right hepatectomy was not referred to as right bi-sectionectomy [11,13]. The term “extended” right hepatectomy was used when there was preservation of the lateral portion of IVa and part of IVb together with the origin of the middle hepatic vein. This was not a formal right trisectionectomy. Similarly, the term “extended” left hepatectomy was used when the right anterior sectoral pedicle was preserved at its origin, but transection was to the right of the middle hepatic vein. Reconstruction was performed with a single Roux loop, but with separate anterior and posterior sectoral anastomoses. Central hepatectomy included resection of IVb and partial V in addition to resection of the extrahepatic biliary tree.

#### 2.4.4. Peri-Operative Outcome

Information on the specific post-hepatectomy complications of haemorrhage, bile leak and liver failure was recorded in compliance with the guidance of the International Study Group for Liver Surgery [14,15,16]. Post-operative complications were also recorded using the system of Dindo–Clavien [17]. Specific detail was also collected on re-admission to hospital within 90 days of surgery and on all-cause post-operative mortality within 90 days of surgery.

#### 2.4.5. Histology

Information was collected on final histopathological status including nodal and margin involvement.

#### 2.4.6. Chemotherapy

Data were collected on the use of chemotherapy with either adjuvant or palliative intent. No patients in this study received peri-operative radiotherapy or neoadjuvant chemotherapy.

#### 2.4.7. Survival

Data were recorded on disease-free survival (DFS), defined as time from surgery until the first radiological evidence of recurrence. Tissue confirmation of recurrent cancer was not required. Data were also recorded on overall post-operative survival (OS), defined as the time from surgery until death.

### 2.5. PH-CCA Work-Up Protocol

Patients in this study were admitted to the service with a clinical diagnosis of PH-CCA based on cross-sectional imaging. All scans were reported by consultant/attending radiologists. Imaging was reviewed at a multidisciplinary tumour board, and patients who had no evidence of metastases on CT or MR and had evidence of a potentially resectable PH-CCA and no co-morbidity precluding surgical resection went forward to detailed assessment. The preparation for surgery algorithm was dichotomised according to the presence or absence of jaundice. Those patients without jaundice who fulfilled the above criteria could proceed directly to surgery. For patients with jaundice, the first intervention step was typically percutaneous trans-hepatic drainage of the future remnant liver. For patients likely to undergo right-sided liver resection, the left hemi-liver was drained first. Bilateral drain placement was only undertaken if drainage of the left hemi-liver resulted in an insufficient fall in bilirubin. Right-lobe biliary drainage was also considered in patients being considered for right portal vein embolisation. For patients likely to require left hemi-liver or extended left-sided resection, the right anterior sectoral pedicle was typically drained first, and the right posterior sector was only drained in those patients with disconnected right anterior and posterior sectoral drainage (type IV PH-CCA). Internal/external drains were the preferred form of drainage as these permitted the external component to be clamped. In addition to relief of jaundice, attention was paid to nutrition, and patients then underwent cardiopulmonary exercise testing [18]. Surgical resection was only undertaken in patients with no evidence of metastatic disease on cross-sectional imaging, with the near-normalisation of serum bilirubin (no greater than 100 µml/L) and satisfactory ventilatory equivalents and anaerobic thresholds on cardiopulmonary exercise testing [18]. Patients who required extended right hepatectomy but had an insufficient left lobe liver volume underwent right portal vein embolisation.

### 2.6. Operative Strategy for PH-CCA Resection

All patients underwent surgery via an open approach. Resectability was confirmed intra-operatively by the confirmation of lack of tumour involvement in future liver inflow assessed with a combination of palpation and dissection. In addition, intra-operative ultrasound was routinely used to exclude the presence of intra-hepatic metastases. Extended lymphadenectomy of stations 12, 8 and 9 was undertaken. The distal extrahepatic bile duct was divided close to the upper border of the pancreas and peeled proximally. The bile duct was then divided proximally prior to hepatectomy. Liver resection was undertaken with the CUSA (Cavitron, Valleylab, Boulder, CO, USA). Partial resection of segment I en bloc with the main specimen was undertaken on a selected basis. Reconstruction was performed with Roux hepaticojejunostomy.

### 2.7. Statistical Analyses

Data are presented as medians (ranges). Overall and disease-free survival was estimated using Kaplan–Meier survival curves and compared using the log-rank test. All statistical analyses were performed using R Foundation Statistical software (R 1.4.1717) with ggplot2 and survival packages (R Foundation for Statistical Computing, Vienna, Austria).

### 2.8. Ethics Approvals

The study was registered as a service evaluation with the Research and Development Department of the Manchester University NHS Foundation Trust (registration number 2977).

## 3. Results

Demographic details and bilirubin profiles on admission and prior to surgery are seen in Table 1. All patients underwent pre-operative contrast-enhanced CT. Twenty-four (75%) underwent pre-operative MR. Five underwent FDG-PET, with all having their scans after 2018. Twenty-three patients underwent pre-operative biliary drainage, with PTC alone being undertaken in thirteen, and two having PTC combined with ERCP). The median (range) number of percutaneous drainage episodes was one (1–5) in those undergoing PTC, and bilateral drainage was required in six.

Twenty-nine patients (91%) had either type III or IV peri-hilar cholangiocarcinoma. All patients with IIIb tumours underwent left hepatic resection. For those with type IV tumours undergoing resection, six (60%) underwent left hepatic resection and three (30%) underwent right trisectionectomy.

### 3.1. Non-Resectional Laparotomy Rate in “No Staging Laparoscopy” Cohort

One patient (3%) in this series underwent a non-resectional laparotomy. This patient had a type IV PH-CCA and had a combination of size-significant nodes along the course of the common hepatic artery and small potential future liver remnant.

### 3.2. Peri-Operative Course

The operative procedures are described in Table 2 and post-operative course in Table 3. Of the two patients (6%) who experienced post-hepatectomy haemorrhage, one required re-laparotomy and under-running of a bleeding point on the hepatic artery, and one was treated with angiographic embolisation. One patient (3%) had post-hepatectomy liver failure after left trisectionectomy. There was no post-operative mortality within 90 days of surgery.

Twenty-nine patients had a final histopathological diagnosis of PH-CCA. One further patient had a final diagnosis of an intraductal papillary neoplasm of the biliary tree (IPNB) with high-grade dysplasia but no invasive cancer.

Nineteen patients (63%) had an involved (R1) resection margin. Eleven had involvement of the liver transection margin, nine the upper bile duct and four the distal duct (one had involvement of both the liver and upper duct margin, one had involvement of liver, upper duct and lower duct, one had liver and lower duct and one had upper and lower duct involvement only).

### 3.3. Chemotherapy

Eleven patients (38%) received chemotherapy after surgery with the patient with a final diagnosis of IPNB without invasive malignancy, the individual with a diagnosis of chronic cholecystitis and the patient with a final diagnosis of metastatic lobular breast carcinoma being excluded. Single-agent capecitabine was used in seven patients, whilst four received a combination of cisplatin/gemcitabine. The median (range) number of cycles of chemotherapy was seven (1–8). The reasons for no adjuvant chemotherapy were as follows: six patients returned to their original referral region and were lost to follow-up, five patients were awaiting the results of an ongoing national study of adjuvant chemotherapy, two did not receive chemotherapy as they were allocated to the observation arm of an adjuvant chemotherapy trial and five had delayed post-operative recovery.

The median (95% CI) time to recurrence was 14 (7–31) months (Figure 1). The median (95% CI) overall survival was 25 (18—upper limit not reached) (Figure 2). The median (95% CI) survival by type was IIIa 22 (8—not achieved), IIIb 23 (12—not achieved) and type IV PH-CCA 25 (11—not achieved) months. Median overall survival (95% CI) for R0 patients was 23 (8—not achieved) and for R1 was 25 (22—not achieved) months (*p* = 0.78; Gehan–Breslow–Wilcoxon).

## 4. Discussion

This study reports the outcome of the surgical management of PH-CCA within a regional hepato-pancreato-biliary service with a work-up algorithm which did not include staging laparoscopy. The infrequency of operable PH-CCA is noted, with 32 patients being regarded as having resectable disease over a 13-year study period. The data contained in this report must be considered in the context of likely sources of bias. This small series is likely to be influenced by selection bias. As the denominator of patients who were considered for surgery but not offered intervention is not available, sampling bias is also likely to influence these results. Data spanning over a decade will entail the use of a range of CT and MR scanners with different resolution and scanning protocols, and thus, reporting bias is likely to introduce further distortion into these results.

Having noted these sources of bias, what then can be learnt from these data? First, accurate pre-operative cross-sectional imaging is critically important. Ideally, imaging should be undertaken prior to biliary intervention. The time interval between cross-sectional imaging and surgery is important, and in this study, the delay from CT to surgery was 31.5 (1–110) days, and the delay from MR to surgery was 28 (1–147) days. If there is a prolonged period of pre-operative biliary drainage, imaging should be repeated prior to surgery. Diagnostic information can also be gained from biliary drainage, and twenty-three (72%) of the patients in this study underwent either PTC or ERCP (with two undergoing both).

In terms of the concordance between pre-operative radiological diagnosis and intra-operative findings, there is broad agreement. Type IIIb PH-CCA diagnosed on pre-operative findings had 100% confirmation intra-operatively. The difference between a diagnosis of type III and IV is small, and thus, there was some difference between pre-operative diagnosis and intra-operative findings (Table 4).

In terms of the type of surgery, the majority of patients underwent hepatic resection combined with excision of the extrahepatic biliary tree. The use of concomitant venous or arterial resection was low in this series.

Setting these data in context, the much larger series of 201 patients from the Amsterdam Medical Center undergoing surgical resection for PH-CCA with a policy of selective diagnostic laparoscopy showed good discrimination between resectable and unresectable disease, with an area under the curve (AUC) on a receiver operator curve of 0.77 (0.68–0.86 95% CI) [19]. However, the same series also reported non-therapeutic laparotomy in 120, with 43 (36%) of these due to locally advanced disease [16]. Intra-operatively discovered peritoneal metastases prevented surgical resection in a further 26 (22%) [19]. These latter data potentially indicate a substantially lower threshold for undertaking surgery, and in such a management algorithm, there may be a useful role for staging laparoscopy.

Detailed assessment of the suitability for surgery with multi-disciplinary team review and the identification of patients who are unfit for surgery because of the presence of metastases, a locally advanced unresectable tumour or co-morbidity represents modern standard of care practice [20,21,22]. At this stage of the work-up pathway there is no indication for staging laparoscopy. The situation becomes more complex in those patients who are thought to be suitable for resection on cross-sectional imaging evidence and who have low co-morbidity. Should laparoscopy be used here as a final diagnostic test prior to surgery, possibly under the same anaesthetic as for resection, and undertaken immediately prior to an open laparotomy? If laparoscopy is used selectively in this fashion, the data presented here firstly do not make a case against such usage, but secondly would indicate that the likely detection of unresectable disease is low. In particular, in this immediate pre-resection use, it does not seem likely that the laparoscopic assessment of the hilus will confirm or deny resectability. In this regard, there are some parallels to the use of staging laparoscopy ± laparoscopic ultrasonography for pancreatic malignancy [23]. This was advocated in the early 2000s but has largely fallen into disuse.

Finally, it is acknowledged that the R1 resection rate of 63% in this series is high (Table 5). Although it could be argued that the more liberal use of a frozen section would have aided the intra-operative recognition of an involved margin, the liver transection was undertaken as far away from the confluence as feasible, and thus, it is unlikely that a more extensive hepatectomy could have been undertaken. Hepatopancreatoduodenectomy was not utilised in this series [24]. Liver transplantation is also an option in this setting and renders R1 resection margins immaterial [25].

In terms of outcomes, this series reports 0% 90-day all-cause post-operative mortality. The median (95% CI) time to recurrence of 14 (7–31) months and the median survival of 25 (18—upper limit not calculable as upper limit not reached) months is acceptable for a series of predominantly type III and IV PH-CCA spanning the era before the introduction of standard post-operative chemotherapy [26].

In summary, this small, single-surgeon cohort of 32 patients undergoing attempted resection for PH-CCA without the use of staging laparoscopy in the work-up algorithm indicates that with careful attention to patient fitness and cross-sectional and interventional radiologic/endoscopic imaging, a very low non-therapeutic laparotomy rate of 3% can be achieved and sustained.

In conclusion, this study demonstrates that the management of rare cancers is difficult, complex and underwritten by a relatively small evidence base, thus highlighting the fact that unexamined dogma should be questioned.

## Figures and Tables

**Figure 1 cancers-14-01841-f001:**
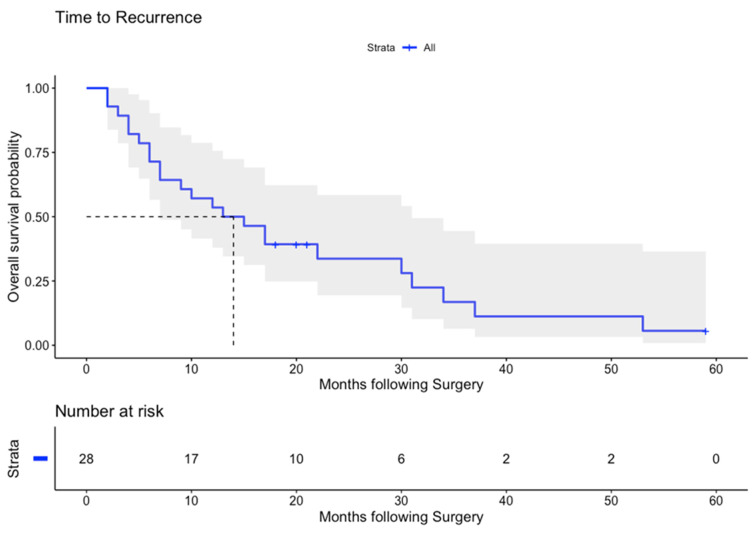
Kaplan–Meier curve for disease-free survival. Data show time to recurrence in months. Shaded blue area shows 95% CI. Number at risk at each timepoint provided.

**Figure 2 cancers-14-01841-f002:**
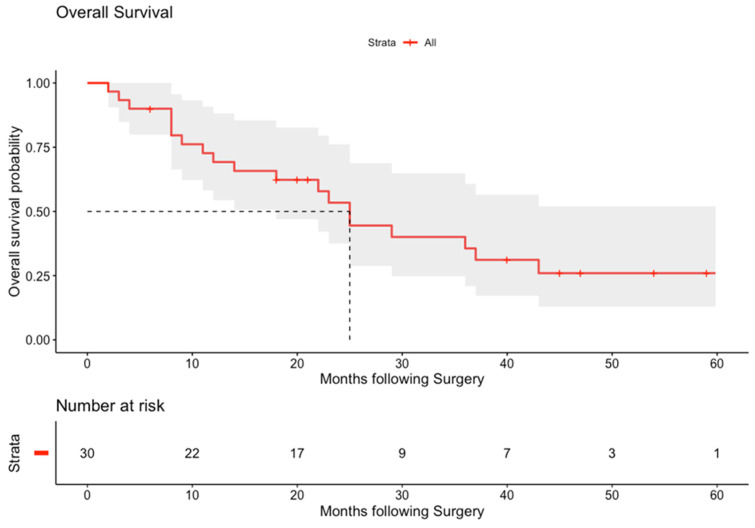
Kaplan–Meier curve for overall survival. Data show overall survival. Shaded blue area shows 95% CI. Number at risk at each time point provided.

**Table 1 cancers-14-01841-t001:** Admission assessment and pre-operative preparation.

	*n* = 32
Female gender (%)	12 (37.5)
Age at operation in years	64 (42–77)
Bilirubin on admission in µmol/L	142.5 (6–397)
Bilirubin prior to surgery in µmol/L	27 (4–163)
Number of patients undergoing resection with bilirubin > 100 µmol/L	3
Delay to normalisation of bilirubin in days (data for 20 patients)	33.5 (0–103)
Delay from index admission to surgery in days	20 (0–89)
Pre-operative CT	32 (100%)
Delay from CT to surgery in days	31.5 (1–110)
Pre-operative MR	24 (75%)
Delay from MR to surgery in days	28 (1–147)
Pre-operative FDG-PET	5 (15.6)
Pre-operative biliary drainage:	
No pre-operative biliary drainage	9 (28)
Pre-operative biliary drainage	23 (72)
PTC	13
ERC	12
Combined PTC/ERC	2

Data are presented as median (range) unless otherwise specified. PTC = percutaneous trans-hepatic cholangiography. ERC = endoscopic retrograde cholangiography.

**Table 2 cancers-14-01841-t002:** Operative detail.

	*n* = 32
Description of operation:	
Right hepatectomy	5
Extended right hepatectomy	2
Right trisectionectomy	3
Left hepatectomy	16
Extended left hepatectomy	1
Left trisectionectomy	1
Extrahepatic biliary tree alone	2
Central hepatectomy	1
Non-resectional laparotomy	1
PH-CCA-specific interventions:	
Excision of extrahepatic biliary tree	30 (94)
Radical hilar lymphadenectomy	30 (94)
Excision of caudate	13 (41)
Portal vein resection and reconstruction	2
Arterial resection and reconstruction	2
Intra-operative frozen section	5
Technical detail:	
Use of Pringle manoeuvre	14 (44)
Median (range) Pringle time in minutes	0 (0–95)
Median (range) transection time in minutes	90 (20–180)
Median (range) operating time in minutes	540 (240–720)
Number requiring intra-operative transfusion	12 (37.5)
Median (range) transfusion requirement in units	0 (0–8)

Types of hepatic resection, concomitant procedures and operative detail are seen. Portal vein reconstruction after resection was with end-to-end anastomosis. Arterial reconstruction after resection was also with end-to-end anastomosis.

**Table 3 cancers-14-01841-t003:** Post-operative course.

	*n* = 32
Specific post-hepatectomy complications:	
Post-hepatectomy haemorrhage	2 (6%)
Post-hepatectomy liver failure	1 (3%)
Post-hepatectomy bile leak	11 (34%)
Clavien–Dindo post-operative complications:	
I	4 (13)
II	6 (19)
IIIa	9 (28)
IIIb	1 (3)
IV	1 (3)
V	0
Median (range) post-operative in-patient stay (in days)	13 (5–44)
Re-admission within 90 days	11 (34)
All-cause post-operative mortality within 90 days	0

Specific post-hepatectomy complications are reported together with an overall Clavien–Dindo categorisation.

**Table 4 cancers-14-01841-t004:** Pre-operative radiological diagnosis, intra-operative correlation and choice of procedure.

PH-CCA Category	Pre-Operative Diagnosis (*n* = 32)	Operative Confirmation	Different Operative Findings	Final Combined Category	Operative Procedure
**II**	4 (13)	3	1 (IIIa)	3 (10)	EBD alone = 2 Central hepatectomy + EBD = 1
**IIIa**	6 (19)	5	1 (IV)	7 (22)	Extended Right hepatectomy = 2 Right hepatectomy = 5
**IIIb**	8 (25)	8	0	12 (37)	Left hepatectomy = 12
**IV**	14 (43)	9	5 (IIIb = 4) (IIIa = 1)	10 (31)	Extended Left hepatectomy = 1 Left hepatectomy = 4 Left trisectionectomy = 1 Right trisectionectomy = 3 Non-resection = 1

Comparison of pre-operative radiological assessment to intra-operative findings and choice of final operative procedure. “Different operative findings” = operative findings are different to the pre-operative radiological diagnosis. “Final combined category” = final assessment including pre-operative radiological diagnosis and operative findings. EBD = extrahepatic biliary tree.

**Table 5 cancers-14-01841-t005:** Histopathology.

	*n* = 32
Median (range) resection specimen weight in gms	365 (167–1565)
Final histopathology:	
Cholangiocarcinoma (PH-CCA)	29
IPNB with high-grade dysplasia	1
Metastatic lobular breast carcinoma	1
Chronic cholecystitis	1
pT, N, M, stage:	
T1, N0	1
T2, N0	10
T2, N1	5
T3, N0	3
T3, N1 *	6
T3, N2	2
T4, N0	1
Cholangiocarcinoma without further detail **	2
IPNB with HGD	1
Chronic cholecystitis	1
R1 resection margin ***	19 (63)

* The patient with a final diagnosis of metastatic lobular breast carcinoma was allocated a pT3, pN1 stage by the reporting histopathologist and is thus included in this category. ** One patient was given a final histologic diagnosis of peri-hilar cholangiocarcinoma without further detail. In addition, the patient who underwent a non-therapeutic laparotomy had biopsy confirmation of the diagnosis of peri-hilar cholangiocarcinoma without allocation of a pTNM stage. *** The denominator for calculation of patients with an R1 resection (*n* = 30) excludes the patient with a non-therapeutic laparotomy and the patient with a final histopathological diagnosis of chronic cholecystitis. IPNB = intraductal papillary neoplasm of the biliary tree. HGD = high-grade dysplasia.

## Data Availability

The data presented in this study are available on request from the corresponding author.

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
