# Peer review of "Work-Up and Outcome of Hepatic Resection for Peri-Hilar Cholangiocarcinoma (PH-CCA) without Staging Laparoscopy"

_cancers, 2022, doi:10.3390/cancers14071841_

Round 1

Reviewer 1 Report

This study reports the outcome of a work-up programme for resection of peri-hilar cholangiocarcinoma (PH-CCA) without the use of staging laparoscopy.
In particular in analysed a single-surgeon cohort of 32 patients undergoing attempted resection for PH-CCA. The study period is 13 years, the cases described are definitely cases of patients with advanced disease, and the survival of the patients is satisfactory. 
Certainly, in such a large study period the case series of only 13 patients is very small, perhaps because they are patients operated on by only one surgeon.   Only one patient (3%) in this series underwent a non-resectional laparotomy and this is definitely the most interesting result.

However, I have serious doubts about the usefulness of these results.

- First of all, the case series is certainly too small to be considered authoritative. Why weren't patients operated on by other surgeons at the center included?
- The authors attribute the low rate of nonresectional laparotomy to an excellent preoperative study, which is undoubtedly essential. However, how many patients were evaluated during the study period and then not proposed for surgery? For what reasons were these patients then not referred for surgery? What were the characteristics that led to the exclusion of these patients?
- The captions of the tables certainly need to be better explained.
- The bibliography needs improvement.

Author Response

Overview

Thank you.  The points made by this reviewer are all fully accepted. In the final analysis, this is indeed a small series.  However, with the surgery of rare cancers, when single-institutional series are examined, all are relatively small. This paper, does however, provide a unique angle in that it queries the dogma of routine laparoscopy.

We have however, fully accepted the reviewer's comments and provide detailed replies to all the points made  and have modified the manuscript accordingly.

  1. First of all, the case series is certainly too small to be considered authoritative. Why weren't patients operated on by other surgeons at the center included?

Reply to point 1:

This is a very important question.  Thank you for this point.  To give a detailed answer, the regional HPB unit in Manchester was established in 2014 by the merger of two units.  Since the merger, the principal surgeon was the sole individual with a published interest in peri-hilar cholangiocarcinoma surgery.  Other individuals operated on a very small number of cases and it is only recently that a formal system of a two-surgeon operating policy has been adopted.  This has been interrupted by COVID and thus the influence of a number of surgeons operating on a small dataset would be considerable and is avoided by focusing on the results of a single individual.  We appreciate that this single-surgeon experience is substantially different to that seen in mainland Europe but it does represent a truthful experience.

  1. The authors attribute the low rate of nonresectional laparotomy to an excellent preoperative study, which is undoubtedly essential. However, how many patients were evaluated during the study period and then not proposed for surgery? For what reasons were these patients then not referred for surgery? What were the characteristics that led to the exclusion of these patients?

Reply to point 2:

Thank you for this important point.  Patients with extra-hepatic disease, hepatic metastases, involvement of arterial or portal inflow to the future remnant liver or ECOG performance status >2 were excluded.  As not all these patients went on to have tissue confirmation of malignancy, it is not possible to state an exact denominator.  However, the text has been modified to provide more detail on the patients who were excluded.

  1. The captions of the tables certainly need to be better explained.

Reply to point 3:

Thank you for this important point.  We have provided additional detail to the captions of all the tables which make them easier to follow.

  1. The bibliography needs improvement.

Reply to Point 4:

Thank you for this point.  We fully accept this.  The bibliography has been updated and enhanced and now features relevant references to adjuvant chemotherapy, liver transplantation and information on the sensitivity of diagnostic laparoscopy.

Reviewer 2 Report

In general well written report of a small, single-surgeon cohort of 32 patients undergoing attempted resection for perihilar cholangiocarcinoma without the use of staging laparoscopy in the work-up algorithm.

As stated by the authors the management of rare cancers is difficult, complex and often based on small evidence. That is completely true. Here they question the use of staging laparoscopy in the work-up. Although I believe the originality and the novelty of the topic is not that big, it can be usefull for surgeons in the field. Again it is an honest and to the point written manuscript.

There are some concerns I would like to address.

  • the numbers are relatively small. Eventhough the numbers are from a highly experienced HPB surgeon, the yearly average is 2-3. Although it is decent addressed in the discussion it remains a small cohort. I question the phrase that the series is unique in modern HPB practice and a suggestion could be to provide additional data from another surgeon/center.
  • the high R1 resection rate should be explained further. What was the correlation of the frozen sections, site of tumor cells left, was there relation with a specific procedure? Additional surgery done in case of distal tumor residu (combined liver resection and pancreatoduodenectomy?)
  • The numbers at risk in figure 1 after 30 months are really too low to say something about, it should be removed
  • to what extent were the vascular resections/reconstructions.
  • I miss numbers on the role of staging laparoscopy and detection of metastases (lnn, peritoneal metastases, liver metastases) in the introduction and would like to see more on the reasons of mandating routine use vs. selective (or no) use. 
  • I think it's worthwhile to elaborate more on the prehabilitation (CPET) of the patients, what cut-off point was used, were people trained upfront or not.
  • how were patients selected for surgery, what was said to be unresectable on imaging?
  • were patients referred for liver transplantation? why (not)? I think the authors should say something about this and what the role of the staging laparoscopy could be. Particularly now the discussion is on for a protocol in GB.

Author Response

Overview

We fully accept the points raised by the reviewer and have modified the manuscript in response.  We now provide more detail in areas such as the R1 rate and a more thorough bibliography including data on the sensitivity of staging laparoscopy.

Point 1:

The numbers are relatively small. Even though the numbers are from a highly experienced HPB surgeon, the yearly average is 2-3. Although it is decent addressed in the discussion it remains a small cohort. I question the phrase that the series is unique in modern HPB practice and a suggestion could be to provide additional data from another surgeon/center.

Reply to point 1:

Thank you for this important point.  Yes, we fully accept it.  We have removed the sentence stating that the series is unique in modern HPB practice as this is un-necessarily overstating the case.  We have also accepted the second component of this reviewer’s first point and agree that future reports would be better if they were multi-institutional and worked to an agreed protocol.

Point 2:

The high R1 resection rate should be explained further. What was the correlation of the frozen sections, site of tumour cells left, was there relation with a specific procedure? Additional surgery done in case of distal tumour residu (combined liver resection and pancreatoduodenectomy?)

Reply to Point 2:

Thank you.  We agree that this is an important point and now provide more detailed explanation.  The distribution of R1 sites is highlighted in the text as follows:  Eleven had involvement of the liver transection margin, 9  the upper bile duct and 4 the distal duct (1 had involvement of both the liver and upper duct margin, 1 had involvement of liver, upper duct and lower duct, 1 had liver and lower duct and 1 had upper and lower duct involvement only).  None of these patients had a positive resection margin on frozen section (although 1 was subsequently positive on formal histopathology).  No patients underwent hepatopancreatoduodenectomy in this series and this is now explained in the text.

Point 3:

The numbers at risk in figure 1 after 30 months are really too low to say something about, it should be removed.

Reply to point 3:

Broadly speaking, the numbers at risk are small at all time points.  Readers are typically more familiar with 5 -Yr survival data and thus, respectfully, we would prefer to retain this.  It also provides complete transparency on outcomes.

Point 4:

To what extent were the vascular resections/reconstructions.

Reply to point 4:

Thank you for this point.  It is also raised by reviewer 1 and technical detail is provided in the caption to the relevant table.

Point 5:

I miss numbers on the role of staging laparoscopy and detection of metastases (lnn, peritoneal metastases, liver metastases) in the introduction and would like to see more on the reasons of mandating routine use vs. selective (or no) use. 

Reply to point 5:

Thanks.  This is a very fair point and the bibliography is now strengthened by a discussion of the sensitivity/specificity of staging laparoscopy.

Point 6:

I think it's worthwhile to elaborate more on the prehabilitation (CPET) of the patients, what cut-off point was used, were people trained upfront or not.

Reply to point 6:

Thank you for this fair point.  We apologise as we may have un-intentionally misled the reviewer.  No patients in this study underwent pre-habilitation. CPET in this context refers to cardio-pulmonary exercise testing.  We have now removed the unexplained abbreviation and replace it with the full phrase.

Point 7:

how were patients selected for surgery, what was said to be unresectable on imaging?

Reply to point 7:

This is an important point and information is now provided in the methods section.

Point 8:

Were patients referred for liver transplantation? why (not)? I think the authors should say something about this and what the role of the staging laparoscopy could be. Particularly now the discussion is on for a protocol in GB.

Reply to point 8:

No patients were referred for liver transplantation as to date this is not available in the United Kingdom.  However, the reviewer makes a very good point about including this in the discussion and we have now modified the discussion to include this.

Round 2

Reviewer 1 Report

The authors have sufficiently revised the manuscript. 

The text now appears sufficient for publication.

Reviewer 2 Report

thank you for taking my points into account. They have been added to the manuscript in a proper way.

no further comments.